# Estimated SARS-CoV-2 Infection and Seroprevalence in Firefighters from a Northeastern Brazilian State: A Cross-Sectional Study

**DOI:** 10.3390/ijerph18158148

**Published:** 2021-08-01

**Authors:** Lysandro P. Borges, Laranda C. Nascimento, Luana Heimfarth, Daniela R. V. Souza, Aline F. Martins, José M. de Rezende Neto, Kezia A. dos Santos, Igor L. S. Matos, Grazielly B. da Invenção, Brenda M. Oliveira, Aryanne A. Santos, Nicolas A. A. Souza, Pamela C. de Jesus, Cliomar A. dos Santos, Marco A. O. Goes, Mércia S. F. de Souza, Adriana G. Guimarães

**Affiliations:** 1Department of Pharmacy, Federal University of Sergipe, São Cristóvão 49100-000, Brazil; lysandro.borges@gmail.com (L.P.B.); larandacarvalho@gmail.com (L.C.N.); luahei@yahoo.com.br (L.H.); Keeziaalves@gmail.com (K.A.d.S.); igor_leonardo100@hotmail.com (I.L.S.M.); graziellyinvencao@hotmail.com (G.B.d.I.); brenda.mascarenhasdossantos71@gmail.com (B.M.O.); ary_anne10@hotmail.com (A.A.S.); nicolasalessandro@gmail.com (N.A.A.S.); pamcjesus@outlook.com (P.C.d.J.); 2Department of Health Education, Federal University of Sergipe, Lagarto 49000-000, Brazil; daniraguer@gmail.com (D.R.V.S.); draalinemartins.orto@gmail.com (A.F.M.); 3Department of Pharmacy, Federal University of Sergipe, Lagarto 49000-000, Brazil; jmkide@gmail.com; 4Sergipe Central Public Health Laboratory, Aracaju 49015-460, Brazil; cliomarsantos@gmail.com; 5Graduate Program in Health Sciences, Federal University of Sergipe, Aracaju 49060-100, Brazil; maogoes@gmail.com (M.A.O.G.); feitosams@gmail.com (M.S.F.d.S.); 6State Health Department, Federal University of Sergipe, Aracaju 49097-670, Brazil

**Keywords:** occupational exposure, epidemiology, firefighters, COVID-19

## Abstract

The new coronavirus has been affecting health worldwide and essential service workers are continually exposed to this infectious agent, increasing the chance of infection and the development of the disease. Thus, this study aimed to estimate the frequency of infection and seroprevalence for SARS-CoV-2 in military firefighters in a city in Northeastern Brazil in January 2021. An observational cross-sectional study was carried out with 123 firefighters who answered a brief questionnaire to collect socio-epidemiological data and underwent RT-PCR and immunofluorescence test (IgM and IgG). The results found reveal a positive seroprevalence, with a high rate of infection in this class of workers, since they are essential service professionals who are exposed to risk due to their working hours, in addition to sharing some spaces and work materials. Besides, there were significant associations between positivity for IgG and IgM, as well as for positive RT-PCR prior to the study and the presence of IgG, with odd ratios of 3.04 and 4.9, respectively. These findings reinforce the need for immunization in this category, whose line of service hinders the adoption of distancing measures, since in many situations physical contact is inevitable.

## 1. Introduction

The infection caused by SARS-Cov-2, called Coronavirus Disease 2019 (COVID-19), corresponds to a type of infectious disease of the respiratory tract that has been recognized as a pandemic by the WHO since the beginning of 2020 (World Health Organization) [1]. As of 31 March 2020, 700,000 COVID-19 cases were confirmed in more than 100 countries and regions and to date, the rapid spread of SARS-CoV-2 has seriously threatened the public health and the economy of most countries around the world [2].

In Brazil, the first case of COVID-19 was notified on 25 February 2020. The subject was a patient who returned to the country after a trip to Italy, but studies by our group suggest that the virus had been circulating in northeastern Brazil one month before [3]. Since then, the disease has spread throughout the Brazilian territory with an increasing number of cases and deaths from the disease in all regions [4]. Currently, Brazil is among the countries with the highest absolute numbers of infections and deaths from COVID-19.

Due to the speed and easy spread of this virus [5], WHO recommends social distancing, in addition to the use of a mask and frequent hand hygiene. However, many workers, especially in essential services, cannot adopt the appropriate social distancing during a pandemic, as is the case of professionals who work in urgent and emergency services, public security and firefighters [6]. As a matter of fact, although the actions of firefighters vary according to state and municipal requirements, as well as doctors and nurses, firefighters are daily exposed to a high risk of contamination [7].

Thus, this study aimed to estimate the frequency of infection and seroprevalence for SARS-CoV-2 in military firefighters in a city in northeastern Brazil in January 2021.

## 2. Materials and Methods

This is an observational cross-sectional study carried out in January 2021, with representatives of the fire brigade of Aracaju, capital of the State of Sergipe, located in the northeastern region of Brazil. The sample, determined in a non-probabilistic manner for convenience, was composed of 123 military firefighters, recruited by the research team as those who are at the forefront in handling cases. The study was approved by the National Bioethics Committee of Brazil (CAAE 31018520.0.0000.5546) and the volunteers who were willing to participate in the study signed a free and informed consent form specific to this research (ICF).

The participants were subjected to a brief questionnaire to collect socio-epidemiological data, where personal data were collected, including age, sex, address, presence of comorbidities and symptoms compatible with COVID-19 (such as fever, dry cough, tiredness, diarrhea, headache, loss of taste or smell, difficulty breathing or shortness of breath, chest pain or pressure, loss of speech or movement) and previous COVID-19 tests. RT-PCR and immunological tests (IgM and IgG) were performed. None of the participants had started the process of immunization against COVID-19 at the time of the survey.

The immunofluorescence assays were carried out in the Department of Pharmacy (Laboratory of Biochemistry and Clinical Immunology, LaBiC-Imm) of the Federal University of Sergipe (UFS). Anti-SARS-CoV-2 IgM and IgG antibodies were detected in the serum using an in vitro diagnostic test system based on lateral flow sandwich detection immunofluorescence technology (Ichroma2™ COVID-19 Ab in conjunction with an Ichroma™ II Reader, Boditech Med Inc., Gangwon-do, South Korea) according to the manufacturer’s instructions. The immunofluorescence method applied showed a sensitivity of 95.8% and specificity of 97%. Concomitantly, after blood collection, all participants gave a nasopharyngeal swab and the samples were sent in a preservative medium to the Central Laboratory of Molecular Biology of the State of Sergipe. Nucleic acid testing was made using an in-house real-time PCR as previously described [8].

The data collected were analyzed using descriptive statistics and expressed as mean, standard deviation and percentage. The associations between the variables studied (positive RT-PCR, seropositivity for IgM and IgG, epidemiological data) were verified using Fisher’s exact test using the GraphPad Prism^®^ 6.0 (GraphPad Software, Inc., San Diego, CA, USA), estimating the odds ratio (OR) and its confidence interval of 95% (95% CI). The level of statistical significance used was *p* < 0.05.

## 3. Results

The sample consisted of 123 fire brigade agents, of which 56 (45.5%) had detectable RT-PCR for SARS-CoV-2, with 17 (13.8%) seropositive for IgM and 32 (26.0%) for IgG. It was possible to verify a significant association between IgM and IgG (*p* = 0.0416, with OR = 3.04 and 95% CI = 1.1–8.7), indicating that firefighters with positive IgM were three times more likely to also have positive IgG.

Most participants were over 40 years old (65.9%), but individuals under the age of 40 had a higher percentage of infection by the virus, in addition to higher rates of seropositivity for IgM and IgG (19% and 33.3%, respectively). There was a predominance of men (99; 80.5%), who also had a higher positive CRP rate (47.5%) and the presence of IgG (26.5%). In addition, most participants lived in the capital of the state where the study was conducted (78.1%), for whom a higher frequency of positive IgM and IgG was also observed (16.7% and 26%, respectively), but with the lowest rate of positive PCR. For any of these variables, a significant association was observed (data not shown).

As for the health profile, only 33.3% (*n* = 41) of the participants had some comorbidity, including such diseases as: cardiovascular (*n* = 16; 39.02%); respiratory (*n* = 13; 31.70%); metabolic, e.g., diabetes and dyslipidemia (*n* = 6; 14.63%); neurological, e.g., migraine, depression, herniated disc (*n* = 3; 7.31%); autoimmune (*n* = 2; 4.87%) and glaucoma (*n* = 1; 2.43%). Among the subgroups of participants with and without comorbidity, there was a small variation between the frequencies of infection and seropositivity (Table 1). Similarly, for these variables there were no significant associations.

As for the presence of general symptoms characteristic of COVID-19, 17.1% (*n* = 21) of the participants reported the presence of at least one of the symptoms during the interview. Of these, 52.4% (*n* = 11) had positive CRP, while only 4.8% (*n* = 1) and 33.3% (*n* = 7) were seropositive for IgM and IgG, respectively. Also, the occurrence of symptoms did not show a significant association with the diagnostic variables, possibly due to the expressive rate of asymptomatic cases (*n* = 45; 80.35%) in this population.

Most participants (*n* = 81; 65.9%) reported not having contact with people who had tested positive or who were asymptomatic. For this group, higher frequencies of PCR (45.7%) and positive IgM (14.8%) were also observed, indicating other means of contamination. The associations tested for this variable were also not significant.

Finally, the participants were asked about the results of previously performed COVID-19 tests (RT-PCR or serologic tests). Twenty-three (18.7%) individuals self-reported positive results for RT-PCR (*n* = 18) or serologic tests (*n* = 5) before participating in the research. Of these, 47.8% (*n* = 11) had positive RT-PCR during the survey, with 17.4% (*n* = 4) and 56.5% (*n* = 13) being seropositive for IgM and IgG, respectively. It is noteworthy that, of these 11 firefighters with a positive RT-PCR test in our study, only one had performed a previous test in a period of less than 2 months, indicating persistent infection. For the other 10, tests had been performed at an interval of 4 to 9 months prior to participation in this study, indicating possible cases of reinfection. There was also a significant association between seropositivity for IgG and previous positive COVID-19 tests (*p* = 0.0064, OR = 4.9% and 95% CI = 1.6–15.2).

## 4. Discussion

The testing of the population, especially of workers in essential services, corresponds to an important strategy for monitoring the evolution and spread of the disease, as well as for the adoption of restrictive measures aimed at controlling its progress. Although other diagnostic methods have been sought [8], RT-PCR (real-time polymerase chain reaction) in respiratory samples is still the gold standard diagnostic test for detecting viral RNA in the host organism [9]. However, the evaluation of serum antibodies, such as IgM and IgG against SARS-CoV-2, is an important method for the study of seroprevalence in different populations over time, since expressive levels of IgM can be detected 10–12 days after the onset of symptoms and IgG antibodies can be detected from the 12th day [10].

In the present study, it was possible to verify that almost half of the participants had detectable PCR for SARS-CoV-2, corroborating the high degree of exposure of these professionals, which consequently has reflected in the number of absences as verified in other studies of Lima et al. [6] and Prezant et al. [11]. In addition, a considerable seropositivity rate was observed for IgM and IgG, with a significant association between the detection of these antibodies, corroborating a previous study recently published [12]. In fact, seropositivity for essential service workers has proven to be of great value, as demonstrated by Melo et al. (2020) [13] in a study carried out with health workers in the same region. 

Most participants live in the capital where the study was conducted, being the main contributors to the observed seroprevalence. These data corroborate Borges et al. [14], who observed a higher prevalence of these antibodies in residents of Aracaju, possibly due to their higher economic concentration and to the current high testing coverage for SARS CoV-2.

About 1/3 of the professionals had some type of comorbidity, with cardiovascular and metabolic diseases being more prevalent. Half of them had a positive result in the RT-PCR test. According to Yang et al. [15] and Liu et al. [16], the most prevalent comorbidities among people infected with COVID are metabolic diseases, and that is the reason why they are classified as patients in the risk group.

Clinically, it was found that most of the firefighters who presented positive PCR were asymptomatic, as it has been observed for the general population [14]. However, such information reinforces the need to comply with the measures recommended by health agencies (use of equipment for individual safety and hand hygiene), since they are professionals whose occupational performance often requires some kind of physical contact.

Another aspect that deserves attention is the high rate of participants who had not had contact with someone who was sick or who had a positive test for the coronavirus (65.9%), which represented about half of those infected and most of those seropositive for IgM. This finding suggests other means of contamination, including the professional activity, which can be due to the long working day (12- or 24-h shifts), in addition to sharing refectory, accommodations and vehicles for long hours, often without presenting the adequate structural and maintenance conditions.

Finally, we found that approximately half of the participants who self-reported positive previous COVID-19 tests showed new positivity for RT-PCR, in addition to high seroprevalence rates for IgM and, especially, IgG. Considering the time between the two evaluations, it is possible that the majority were cases of reinfection. The agreement of these data can be observed through the statistically significant association between the previous positive tests report and the detection of IgG, with OR = 4.9, also corroborating the immune response against exposure to SARS-CoV-2 reported in literature [17,18]. Similarly, other studies have reported cases of reinfection in essential service workers such as health professionals [19,20]. It is also worth highlighting the occurrence of the Brazilian variant P1 in the period of development of this study, further reinforcing the possibility of reinvention by SARS-CoV-2 [21].

These findings reinforce the need for the immunization of this category of workers who, being essential, present extensive work schedules, collective work, often in restricted spaces and work activity that normally requires physical contact (e.g., rescues), making them constantly exposed to the corona virus.

## 5. Conclusions

Through this study, it was possible to verify that firefighters have high exposure to the corona virus, which can be seen by the considerable rate of infection and seroprevalence for IgM and IgG. These findings reinforce the need for immunization in this category, whose line of service hinders the adoption of distancing measures, since in many situations physical contact is inevitable.

## Figures and Tables

**Table 1 ijerph-18-08148-t001:** Demographic and health profile of the firefighters participating in the study.

Sociodemographic Variables	Total		RT-PCR Positive		SARS-CoV-2 IgM Seropositivity		SARS-CoV-2 IgG Seropositivity	
	*n*	%	*n*	%	*n*	%	*n*	%
Firefighter	123	100	56	45.5	17	13.8	32	26.0%
Age Group (41.6 ± 9.6)								
≤40 years old	42	34.1	21	50	8	19.0	14	33.3
>40 years old	81	65.9	35	43.2	9	11.1	18	22.2
Gender								
Male	99	80.5	47	47.5	12	12.1	26	26.3
Female	24	19.5	9	37.5	5	20.8	6	25.0
Locality								
Capital	96	78.1	43	44.8	16	16.7	25	26.0
Other municipalities	27	8.9	13	48.1	1	3.7	7	25.9
Comorbidities								
No	82	66.7	34	41.5	9	11.0	9	28.0
Yes	41	33.3	22	53.7	8	19.5	23	22.0
COVID Symptoms								
No	102	82.9	45	44.1	16	15.7	25	24.5
Yes	21	17.1	11	52.4	1	4.8	7	33.3
Contact with Infected/Symptomatic Person								
No	81	65.9	37	45.7	12	14.8	17	21.0
Yes	42	34.1	19	45.2	5	11.9	15	35.7
Previous Positive COVID-19 Tests Self-Report	23	18.7	11	47.8	4	17.4	13	56.5

## Data Availability

Data are available and can be obtained from the correspondence author (Adriana Guimarães, adrianagibara@academico.ufs.br).

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
