# Peer review of "Estimated SARS-CoV-2 Infection and Seroprevalence in Firefighters from a Northeastern Brazilian State: A Cross-Sectional Study"

_ijerph, 2021, doi:10.3390/ijerph18158148_

Round 1

Reviewer 1 Report

This study aimed to estimate the frequency of infection and seroprevalence for SARS-CoV-2 in military firefighters in a city in northeastern Brazil in January 2021. An observational cross-sectional study was conducted with 123 firefighters who responded to a short questionnaire to collect socio-epidemiological information and undergo RT-PCR and immunofluorescence tests (IgM and IgG). The results revealed positive seroprevalence, with a high rate of infection in this class of workers. These findings reinforce the need for immunization in this category, whose line of service hinders the adoption of distancing measures as physical contact is unavoidable in many situations. This study showed an interesting report, and it pays attention to a class of workers who could not comply with the safety regulations provided in the COVID period, receiving a real statistic of the COVID-19 infection. The text is clear, the results and discussions are amply sufficient to describe the work done.

Author Response

Dear reviewer, we are glad that you enjoyed the work. All requested adjustments were made, including an English revision.

Reviewer 2 Report

The manuscript describes the incidence of SARS-CoV-2 infection in a population of 123 firefighters, who are a particularly at risk population due to their job. In my opinion, besides an overall re-writing to improve the English form, some points need clarification. More in detail:

1) Results, line 83: please better eplain the correlation between IgM and IgG levels

2 ) Table 1: the sum of subjects with and without symptom is < than 100%

3) Results, line 113-118: It appears that 18.7% had a positive RT-PCR obtained before theenrollment in the study: I guess that it could be of interest to know the interval elapsinge between these 2 different investigation in order to separate a persisting infection from a true  reinfection.

Author Response

Dear Reviewer, thank you for your comments. All requested adjustments were made, including an English revision.

1) Results, line 83: please better explain the correlation between IgM and IgG levels

It was possible to verify a significant association between IgM and IgG (p = 0.0416, with OR = 3.04 and 95% CI = 1.1 - 8.7), indicating that firefighters with positive IgM were three times more likely to also have positive IgG.

2) Table 1: the sum of subjects with and without symptom is < than 100%

Sorry. I did not understand. There were 102 participants without symptoms (82.9%) and 21 with symptoms (17.1%).

3) Results, line 113-118: It appears that 18.7% had a positive RT-PCR obtained before theenrollment in the study: I guess that it could be of interest to know the interval elapsinge between these 2 different investigation in order to separate a persisting infection from a true reinfection.

The information was inserted in the results:

 It is noteworthy that, of these 11 firefighters with a positive RT-PCR test in our study, only 1 had performed a previous test in a period of less than 2 months, indicating persistent infection. For the other 10, tests had been performed at an interval of 4 to 9 months prior to participation in this study, indicating cases of reinfection. The discussion has also been improved.

Reviewer 3 Report

Dear Authors, thank you for giving me the opportunity to read your work. It explores the positivity for COVID-19 with RT-PCR or IgG/IgG tests among firefighters.

Like many others, they get often in contact with people that could be positive for COVID-19. In this way, I did not find so novel your work. In addition, I doubt it would be found of high interest for the readers.

I have also some questions:

How many fire-fighters are in the fire brigade of Aracaju? 

page 3 line 93 --> data not shown: why?

page 3, table 1: mean age 61.5?? Is it correct? Mean+ SD=88 years? I think this is an error. (I hope so).

Age group (61.5 ± 27.5) 

Page 3 line 103: please define symptoms of COVID

page 4 line 112: data not shown: why?

Did someone get vaccinated against COVID-19 before the study?

Discussion: near half of the the participants had detectable PCR for SARS-CoV-2, corroborating the high degree of exposure of these professionals....--> what type of DPI were available at the time of the study? Any of the participants used them? Did they receive instructions about how to correctly use DPI?

What type of serologic test did you perform? CLIA against spike protein of SARS-COV2? Other?

I suggest also English revision.

Best regards 

Author Response

Dear reviewer, we respect your opinion, but we emphasize that firefighters, as well as other essential service workers, continued to carry out their activities throughout the pandemic (including saving lives) and, even continually exposed to the risk of infection, had no priority in immunization in many countries. This study draws attention to this issue. Still, all requested adjustments were carried out as follows below. Thank you for the contributions.

I send the requested information below:

1. How many fire-fighters are in the fire brigade of Aracaju? 

We have approximately 520 firefighters.

2. page 3 line 93 --> data not shown: why?

Associations between the parameters described in the paragraph were tested, but as they did not indicate statistical relevance, we chose not to show these data.

3. page 3, table 1: mean age 61.5?? Is it correct? Mean+ SD=88 years? I think this is an error. (I hope so).

Age group (61.5 ± 27.5) 

Thank you for alerting us, the correct values are: 41.6 ± 9.6.

4. Page 3 line 103: please define symptoms of COVID

Symptoms of covid-19 ranged from mild to severe, as described below:

- Most common symptoms: fever, dry cough, tiredness;

- Less common symptoms: aches and pains, sore throat, diarrhea, conjunctivitis, headache, loss of taste or smell, skin rash or discoloration of fingers or toes;

- Severe symptoms: difficulty breathing or shortness of breath, chest pain or pressure, loss of speech or movement.

This information has been added to the methods.

5. page 4 line 112: data not shown: why?

The associations tested for this variable were also not significant.

6. Did someone get vaccinated against COVID-19 before the study?

The immunization in Aracaju began on January 19, 2020 and did not include the group of military firefighters. Still, they were all asked about the possibility of having received a dose of vaccine, but all participants reported not having received it.

This information has been added to the methods.

7. Discussion: near half of the the participants had detectable PCR for SARS-CoV-2, corroborating the high degree of exposure of these professionals....--> what type of DPI were available at the time of the study? Any of the participants used them? Did they receive instructions about how to correctly use DPI?

Yes, it is part of the internal security protocol to use either surgical or N95 mask and procedure gloves, in addition to protective glasses and disposable apron.

8. What type of serologic test did you perform? CLIA against spike protein of SARS-COV2? Other?

We performed immunofluorescence assays for Anti-SARS-CoV-2 IgM and IgG antibodies.

9. I suggest also English revision.

The English was revised.

Round 2

Reviewer 2 Report

All the suggestions about the previous forma have been fulfilled

Author Response

Thanks for your Round 1 suggestions and for appreciating our work so carefully. 

Reviewer 3 Report

As it regards the submission in the object, during first round I decided to reject (please see my comments in the previous review). I continue to think that this paper does not add anything of value to the scientific community, but overall I believe it is not sufficiently suitable for publication in your esteemed journal.

Author Response

Dear evaluator, we once again respect your opinion, but we reiterate the importance of our work, since it leads to reflection on the continuous exposure of workers in public security services. There was no incentive for testing these workers and our study demonstrates that part of those who tested positive for COVID-19 were asymptomatic, being therefore a source of transmission to colleagues and society. Another aspect is that, unfortunately, the Brazil took a long time to acquire vaccines, slowing down the immunization process and exposing professionals for a long time, in addition to several deaths in this category. For all these aspects, I defend, on behalf of other authors, the importance of this work for the scientific community and for the area of ​​epidemiology. Respectfully, 

Dr Adriana Gibara Guimarães